# Comparative Evaluation of the Cytotoxic Effects of Metal Oxide and Metalloid Oxide Nanoparticles: An Experimental Study

**DOI:** 10.3390/ijms24098383

**Published:** 2023-05-06

**Authors:** Marina P. Sutunkova, Svetlana V. Klinova, Yuliya V. Ryabova, Anastasiya V. Tazhigulova, Ilzira A. Minigalieva, Lada V. Shabardina, Svetlana N. Solovyeva, Tatiana V. Bushueva, Larisa I. Privalova

**Affiliations:** Yekaterinburg Medical Research Center for Prophylaxis and Health Protection in Industrial Workers, 30 Popov Street, 620014 Yekaterinburg, Russia

**Keywords:** nanoparticles, cytotoxicity, BALF, intratracheal instillation, rats

## Abstract

Industrial production generates aerosols of complex composition, including an ultrafine fraction. This is typical for mining and metallurgical industries, welding processes, and the production and recycling of electronics, batteries, etc. Since nano-sized particles are the most dangerous component of inhaled air, in this study we aimed to establish the impact of the chemical nature and dose of nanoparticles on their cytotoxicity. Suspensions of CuO, PbO, CdO, Fe_2_O_3_, NiO, SiO_2_, Mn_3_O_4_, and SeO nanoparticles were obtained by laser ablation. The experiments were conducted on outbred female albino rats. We carried out four series of a single intratracheal instillation of nanoparticles of different chemical natures at doses ranging from 0.2 to 0.5 mg per animal. Bronchoalveolar lavage was taken 24 h after the injection to assess its cytological and biochemical parameters. At a dose of 0.5 mg per animal, cytotoxicity in the series of nanoparticles changed as follows (in decreasing order): CuO NPs > PbO NPs > CdO NPs > NiO NPs > SiO_2_ NPs > Fe_2_O_3_ NPs. At a lower dose of 0.25 mg per animal, we observed a different pattern of cytotoxicity of the element oxides under study: NiO NPs > Mn_3_O_4_ NPs > CuO NPs > SeO NPs. We established that the cytotoxicity increased non-linearly with the increase in the dose of nanoparticles of the same chemical element (from 0 to 0.5 mg per animal). An increase in the levels of intracellular enzymes (amylase, AST, ALT, LDH) in the supernatant of the bronchoalveolar lavage fluid indicated a cytotoxic effect of nanoparticles. Thus, alterations in the cytological parameters of the bronchoalveolar lavage and the biochemical characteristics of the supernatant can be used to predict the danger of new nanomaterials based on their comparative assessment with the available tested samples of nanoparticles.

## 1. Introduction

Particles in the nanometer range are considered to be an airborne hazard, mainly because of their size, which determines their interaction with the structures of an organism [1]. Larger particles tend to deposit in the upper airways, smaller ones deposit in the trachea and bronchi, while fine particles (<2 μm) can be delivered to the alveoli and are more prone to systemic action [2]. Metal nanoparticles (NPs), for instance, exhibit the highest systemic toxicity and to a greater extent cause local reactions in the organism following inhalation exposure [3], some of which are determined by the specific characteristics of the chemical element that form the nanoparticles and activate certain mechanisms of toxicity. It is also believed that the cytotoxicity of nanoparticles increases with the atomic number of their basic chemical element [4].

Numerous studies both in vitro and in vivo have demonstrated the cytotoxic effects of nanoparticles on lung tissue. A stimulation of reactive oxygen species production and a decrease in the level of adenosine triphosphate was observed in the culture of the A549 cancer cell line obtained from a human bronchoalveolar carcinoma following exposure to aluminum, titanium, and silicon dioxide nanoparticles sized 10 to 60 nm [5]. The exposure to silicon dioxide nanoparticles with diameters of 15 and 46 nm induced a dose-dependent decrease in cell viability and a simultaneous increase in the concentration of reactive oxygen species (ROS) in the medium [6]. Zinc oxide nanoparticles had a toxic effect on human pulmonary alveolar epithelial cells (HPAEpiC), inducing ROS formation and accumulation in mitochondria by inhibiting the activity of superoxide dismutase (SOD) and reducing the level of glutathione (GSH). ROS, in turn, open the mitochondrial Ca^2+^ pathway and reduce mitochondrial membrane potentials, leading to apoptosis [7]. Yu et al. [8] also reported the influence of zinc oxide NPs on the membrane potential of mitochondria. A comparative analysis of the effects of Fe_2_O_3_ and ZnO NPs on BEAS-2B and A549 cell lines showed that ZnO NPs, but not Fe_2_O_3_ NPs, cause cell cycle arrest, cell apoptosis, ROS production, mitochondrial dysfunction, and impaired glucose metabolism, which are responsible for cytotoxicity [9]. Fe_3_O_4_ NPs, however, can also lead to disturbances in mitochondrial activity, boost ROS production in cells, and lead to a significant decrease in the ATP level [10].

Experiments in vivo have shown a toxic effect of indium tin oxide nanoparticles in rats inducing acute inflammation after intratracheal instillation [11]. A short-term inhalation exposure to cerium oxide nanoparticles caused pulmonary inflammation in rats [12]. An intratracheal instillation of CuO NPs in C57BL/6 mice caused lung fibrosis by inducing apoptosis of epithelial cells, partially attributed to an increased number of ROS, and exacerbating inflammation in the lung tissue in a dose-dependent manner [13]. An experiment on rats showed that a 10-month chronic inhalation exposure to NiO NPs induced manifestations of systemic toxicity against unexpressed lung pathology related to a rather low chronic retention of nanoparticles in lungs [14]. Dumkova et al. [15] demonstrated a similar systemic toxicity following inhalation exposure to lead oxide nanoparticles.

Intratracheal instillations are just as useful for ranking the toxicity of nanoparticles as inhalation exposure studies [16]. It is, therefore, of interest to study the cytotoxicity of nanoparticles of different chemical natures instilled intratracheally at various doses.

## 2. Results

### 2.1. Comparative Analysis of Nanoparticles of Various Chemical Natures

#### 2.1.1. Cytological Characteristics of Bronchoalveolar Lavage Fluid (BALF) Following Exposure to Nanoparticles

In the series of nanoparticles of various element oxides (CuO, PbO, CdO, Fe_2_O_3_, NiO, and SiO_2_) tested at a dose of 0.5 mg per animal, we established that CuO and CdO NPs caused the greatest influx of cells into the lungs, at a rate that was 8.4 and 6.0 times higher compared to the controls, respectively (*p* < 0.05) (Figure 1a). The smallest 1.3 and 1.8-fold influxes were observed following exposure to Fe_2_O_3_ and SiO_2_ NPs, respectively.

The exposure to CuO NPs was found to induce the highest increase in the number of neutrophilic leukocytes (by 34.9 times compared to the controls, *p* < 0.05) followed by the exposure to CdO NPs, leading to a 22.3-fold increase. Instilled Fe_2_O_3_ and SiO_2_ nanoparticles, in their turn, did not affect this indicator.

A statistical increase in the number of alveolar macrophages was established following the exposure to NiO nanoparticles (by 1.85 times, *p* < 0.05). We found an increase in the number of alveolar macrophages to approximately the same degree after the exposure to CdO and CuO nanoparticles (by 1.64 and 1.86 times, respectively).

The main indicator of a cytotoxic effect is the ratio of neutrophilic leukocytes to alveolar macrophages. Its value demonstrated the highest (17.7-fold) increase under the effect of CuO NPs and a slightly lower one (by 14.3 times) following the exposure to CdO NPs (*p* < 0.05). The smallest changes in the ratio were observed in the rats exposed to Fe_2_O_3_ (by 1.8 times compared to the controls, *p* < 0.05) and SiO_2_ nanoparticles.

In the series of nanoparticles of element oxides (NiO, Mn_3_O_4_, SeO, and CuO) tested at a dose of 0.25 mg per animal (Figure 2), the largest influx of cells to the lungs was found in the NiO NP exposure group, which appeared to be 7.7 times higher than in the controls (*p* < 0.05), owing to a 77.5-fold increase in the number of neutrophilic leukocytes and a 3.6-fold increase in that of alveolar macrophages (*p* < 0.05).

The Mn_3_O_4_ NPs increased the influx of neutrophilic leukocytes by 26.3 times compared to the controls (*p* < 0.05). The dose of 0.25 mg CuO NPs per animal induced a significant influx of neutrophilic leukocytes (by 3.0 times compared to the controls, *p* < 0.05).

The NiO, Mn_3_O_4_, and CuO nanoparticles demonstrated a pronounced cytotoxic effect, as shown by the NL/AM ratios, which were 20, 11.2, and 2.1 times higher than that in the controls (*p* < 0.05). It is worth noting that the SeO NPs caused a slight increase in the total BALF cell count (by 1.5 times compared to the controls, *p* < 0.05), while other BALF cytological parameters only tended to increase.

#### 2.1.2. Biochemical Parameters of BALF Following Nanoparticle Exposure

Altered biochemical indices of the supernatant of the bronchoalveolar lavage were found after the exposure to NPs at a dose of 0.5 mg per animal (Figure 3). The highest increases in the level of enzymes under study (amylase, ALT, AST, GGT, and LDH) were noted following the exposures to CdO and CuO NPs. Mild reactions were observed following the instillations of PbO, NiO, and SiO_2_ NPs. Fe_2_O_3_ NPs, on the opposite, led to a decrease in the levels of enzymes in the BALF supernatant.

The exposure to element oxide nanoparticles at a dose of 0.25 mg per animal (Figure 4) caused changes in enzyme activity to one degree or another compared with the controls.

A significant increase in enzyme activity in the supernatant was observed by the level of amylase and LDH after exposure to Mn_3_O_4_ NPs. The LGH level was also found to be higher in the CuO NP exposure group.

### 2.2. Comparative Analysis of Nanoparticles at Different Exposure Doses

#### 2.2.1. Characteristics of BALF after Intratracheal Instillation of Copper Oxide Nanoparticles

The cytological characteristics of BALF after the instillation of CuO NPs at different doses did not change monotonic with the increasing dose (Figure 5).

The greatest response was observed to a dose of 0.5 mg per animal. At the same time, a lower dose of 0.2 mg per animal caused greater changes in the influx of neutrophilic leukocytes than that of 0.25 mg per animal. This led to a statistical increase in the NL/AM ratio after the administration of CuO NPs at a dose of 0.2 mg per animal, while that of 0.25 mg per animal only indicated a rising trend.

The values of the biochemical parameters of the supernatant after the instillation of CuO NPs rose monotonically with the NP dose (Figure 6). The most indicative in this regard were such enzymes as ALT, AST, and LDH.

#### 2.2.2. Characteristics of BALF after Intratracheal Instillation of Lead Oxide Nanoparticles

The indicator of the total BALF cellularity demonstrated a monotonic dose-dependent increase (Figure 7a). At the same time, the influx of neutrophilic leukocytes and alveolar macrophages was higher at a lower dose of NPs. Hence, the main cytotoxicity index (NL/AM ratio) was also higher in response to a lower dose of PbO NPs.

The pattern of alterations in the biochemical parameters of the supernatant with an increase in the dose of PbO NPs was ambiguous (Figure 8). Only the GGT level increased statistically in response to the dose of 0.2 mg per animal. The remaining indices showed a tendency either to a monotonic increase (like amylase) or to a monotonic decrease (ALT), changed non-monotonically (AST), or remained at the level of control values (LDH).

#### 2.2.3. Characteristics of BALF after Intratracheal Instillation of Nickel Oxide Nanoparticles

We observed a statistical increase in all the studied cytological characteristics of BALF after the exposure to NiO NPs at a dose of 0.25 mg/rat (Figure 9). A higher dose of 0.5 mg/rat induced an increase in the total BALF cellularity (*p* < 0.05), as well as a statistically significant increase in the number of neutrophils (*p* < 0.05). It is clearly seen that the cytological changes were more pronounced following the lower-dose exposure.

The studied biochemical parameters tended to increase following the exposure to NiO NPs at different doses (Figure 10).

## 3. Discussion

### 3.1. Relationship between the Chemical Composition of Nanoparticles and Their Cytotoxicity

When foreign particles, including nanoparticles, enter the lower airways, the phagocytic activity is mobilized, which is expressed by an increase in the number of neutrophilic leukocytes (NL) and alveolar macrophages (AM) in the bronchoalveolar lavage fluid (BALF). Alveolar macrophages are the first to phagocytize foreign particles. The products of their destruction stimulate the influx of neutrophilic leukocytes, and the more cytotoxic these particles are, that is, the more damage they cause in the lung tissue or the population of phagocytic cells, the greater the influx of neutrophilic leukocytes [17,18]. The cellular shift towards neutrophilic leukocytes, which is estimated by the NL/AM ratio, is therefore a key indicator of the cytotoxic effect [19,20,21].

When assessing the impact of the chemical nature of nanoparticles at a dose of 0.5 mg per animal, the largest neutrophilic leukocyte influx was observed in the CuO NP exposure group (Figure 2a). It is important to note that, among the elements studied in the form of oxide nanoparticles, copper is an essential element, while the CuO NPs had an even greater cytotoxic effect than the particles of such non-essential elements as lead and cadmium. This might be attributed to the fact that CuO NPs are highly redox active nanoparticles leading to high ROS formation due to their large active surface area, rapid dissolution, and an easy change in the oxidation state of dissolved copper [22]. Other authors also reported higher cytotoxicity and DNA damage due to CuO NPs compared to TiO_2_, ZnO, CuZnFe_2_O_4_, Fe_3_O_4_, and Fe_2_O_3_ nanoparticles, carbon and nanotubes [23]. In a recent review [24] on the effect of copper nanoparticles on the lungs of mice and rats following intranasal exposure, the authors mentioned the development of inflammation, ROS generation, alveolitis, bronchiolitis, fibrosis, and destruction of the epithelium. The intratracheal instillation of CuO NPs induced acute bronchioloalveolar inflammation with diffuse pulmonary edema, indicating the pulmonary toxicity of CuO NPs [25]. An excess of free copper ions leads to a cascade of redox reactions leading to formation of reactive oxygen species, which destroy the cell both inside and outside. We must note that the ROS generation additionally induced by membrane lipid peroxidation under effect of NPs can lead to a loss of membrane elasticity, which, like an abnormally high fluidity, inevitably leads to cell death [1]. In addition, with regard to such essential elements as copper, it is impossible to exclude the contribution of mechanisms intended for their entry into the cell from the implementation of the toxic effect. In their experiments on HepG2 hepatocytes, Cuillel et al. [26] showed that NPs penetrate inside (most likely by endocytosis), bypassing the cellular defense mechanisms against excess copper.

The PbO NPs had a cytotoxic effect, as shown by the increase in total cellularity and the NL/AM ratio (Figure 2a,d), but to a much lesser extent than CuO or CdO NPs, which is most likely related to the lower cytotoxicity of this element. It is known that semi-lethal doses of lead and its compounds are higher than those of cadmium [27].

The NiO NPs at a dose of 0.5 mg per animal caused the largest influx of alveolar macrophages compared to the other NPs (Figure 2c). At the same time, at a lower dose, the NiO NPs caused a steady increase in all the cytological parameters of BALF (Figure 3). A neutrophilic and lymphocytic inflammatory response appeared 24 h after intratracheal instillation of NiO NPs in a study by Jeong et al. [28]. Other researchers have demonstrated that nickel NPs induce oxidative stress, severe and persistent inflammation in the lungs, and fibrosis [29,30].

The instillation of Mn_3_O_4_ NPs increased the influx of neutrophilic leukocytes and the NL/AM ratio revealing the cytotoxicity of those NPs. Manganese is essential for living organisms as a cofactor for enzymes, yet, high levels of environmental (polluted water) and occupational (welding) exposure to this metal can lead to neurological disorders, including manganism, a condition similar to Parkinson’s disease. A recent review has presented data on the toxic properties of manganese and its authors conclude that Mn NPs can be both toxic and non-toxic. The latter is observed with various modifications of NPs [31], e.g., BSA-modified Mn_3_O_4_ NPs exhibit excellent antioxidant activity [32].

A number of studies have shown that iron NPs have a low pulmonary toxicity both in vitro and in vivo [33,34,35]. In our experiment, the Fe_2_O_3_ NPs caused a minimal influx of neutrophilic leukocytes and a decrease in the number of alveolar macrophages, leading to a significant increase in their ratio. It is noteworthy that in this case too, the NL/AM ratio indicates the cytotoxic effect of Fe_2_O_3_ NPs, although it does not cause an active macrophage influx into the lungs. The latter may indirectly indicate a lesser destruction of cells that have absorbed Fe_2_O_3_ NPs compared to the effect of other metal oxide NPs.

The SiO_2_ and SeO NPs demonstrated no pronounced cytotoxic properties in our study. For SiO_2_ NPs, this was probably due to the inertness and low solubility of this oxide in biological fluids. Fukui et al. [36] showed that the contribution of oxidative stress to the pulmonary toxicity of crystalline SiO_2_ was minimal at the early acute stage after exposure. Toxic effects have been observed following intratracheal administration of synthetic amorphous silica nanomaterials, but the genotoxic effects of SiO_2_ NPs were null [37]. Only a significantly higher dose of SiO NPs (50 mg/kg) with a longer exposure (7 to 56 days) has led to the development of granulomatous inflammation in the lungs of rats, followed by progressive and massive fibrous nodules [38].

In the case of SeO NPs, we can speak about the manifestation of the dual nature of selenium: the cytotoxic effect of its nanoparticles on the one hand and the protective effect of selenium as a trace element on the other. The NPs are insoluble in water and highly soluble in biological fluids [39], thus suggesting the incomplete dissolution of SeO NPs in the upper airways. Those undissolved particles could contribute to an increase in the influx of cells due to some mobilization of neutrophilic leukocytes and alveolar macrophages, as evidenced by their slightly increased ratio. It was these particles deposited in the respiratory tract that phagocytized alveolar macrophages: Chen et al. [40] argue that selenium nanoparticles themselves are the “driving force” of toxicity, while dissolved ions make only a minor contribution. At the same time, alveolar macrophages were not yet damaged by the absorbed particles and were not destroyed, which can be judged by the biochemical parameters of the BALF supernatant, in which the content of the intracellular enzymes did not differ statistically from the corresponding control values. We suggest that selenium, as an antioxidant, is involved in the inhibition of oxidative stress induced by insoluble NPs at its various levels.

The biochemical testing of the supernatant deepened the analysis of our experimental data.

In our studies, GGT showed no pronounced sensitivity to the instillation of nanoparticles. Despite the apparent cell destruction, the natural variability of this enzyme is likely to be high, which led to a large scatter of the data and the absence of significant differences. Interestingly, GGT is a membrane-bound enzyme involved in the transfer of the glutamyl moiety of glutathione to other amino acids and dipeptides [41].

The intracellular glycolytic enzyme LDH is used to analyze the integrity of the cell membrane [25,42,43]. It is an important marker of cellular destruction. A high degree of cell membrane destruction occurred after exposure to the CuO NPs (at doses of 0.25 and 0.5 mg per animal), CdO NPs (at a dose of 0.5 mg per animal), and Mn_3_O_4_ NPs (at a dose of 0.25 mg per animal), as can be seen from Figure 3e and Figure 4c. The nanoparticles of other element oxides did not lead to significant increases in LDH.

Increases in amylase were observed after the instillation of the same CuO, CdO, and Mn_3_O_4_ NPs, probably associated with a large influx of cells into the lungs and their destruction. Cell death causes a release of fibrogenic factors, which lead to the activation of the growth of connective tissue. An increase in the level of amylase in BALF has been shown in inflammatory and carcinogenic lesions of the lungs [44,45,46], as well as in the exacerbation of chronic obstructive pulmonary disease [47] and aspiration pneumonia [48,49]. All the above processes are accompanied by inflammation and cell proliferation, be it tumor cells or fibroblasts. We assume that it is the active growth of the connective tissue that accounts for a rise in amylase in the supernatant and can serve as a characteristic not of cytotoxicity, but of fibrogenicity of the particles in question.

Exposure to the same CuO, CdO, and Mn_3_O_4_ NPs led to an increase in AST and ALT levels in the supernatant. These two enzymes are also intracellular (mainly located in the cytosol) [50], thus indicating damage to lung cells as well. Therefore, an increase in intracellular enzymes in the supernatant indicates an explicit cytotoxic effect of NPs.

### 3.2. Relationship between the Dose of Nanoparticles and Their Cytotoxic Effects

It is known that the vast majority (85.7%) of dose–response relationships observed after exposure to various pollutants are nonlinear [51]. Such polyphase links have been repeatedly shown in in vitro experiments [52,53]. Thus, according to the main cytological indicator, i.e., the ratio of neutrophilic leukocytes to alveolar macrophages, the highest response following exposure to PbO NPs (Figure 8d) occurred at a dose of 0.2 mg per rat, which was not the highest among those tested by us. At the same time, a similar trend can be traced for some biochemical parameters of the supernatant (AST, GGT) (Figure 9c,d). The contribution of dimensionality cannot be ruled out either, e.g., particles 47 ± 16 nm in size at a dose of 0.2 mg per animal (in contrast to smaller particles 23 ± 5 nm in size at a dose of 0.5 mg per animal) could be easily recognized by defense systems of the body [1] and phagocytosed without causing damage to lung cells, alveolar macrophages, or neutrophilic leukocytes.

In the studies of exposure to NiO NPs at different doses, similar to the PbO NPs, changes in the cytological parameters in response to increasing doses were non-monotonic (Figure 9). In contrast to the PbO NPs, the NiO NPs were all of the same size. The main indicator of nanoparticle cytotoxicity was statistically increased after the NiO NP exposure at the dose of 0.25 mg/rat due to an active influx of neutrophils while it only tended to increase under the effect of a higher exposure level. It can be assumed that the NP dose contributes more to the toxic effect of nanoparticles than their size but this assumption still needs to be tested. At the same time, the levels of enzymes, including intracellular ones (Figure 10), in the supernatant were somewhat increased, which may indirectly indicate the destruction of some cells under the influence of trapped NPs.

The exposure to CuO NPs induced the highest response at the dose of 0.5 mg per rat, followed by that of 0.2 mg but not of 0.25 mg per rat (Figure 6). At the same time, judging by the evaluation of the biochemical parameters of the bronchoalveolar lavage fluid (Figure 7), we can assume a linear dose–effect relationship.

An increase in the influx of cells owing to alveolar macrophages and, mainly, neutrophilic leukocytes (Figure 6), which does not correlate with changes in BALF biochemical parameters (Figure 7), can be attributed to a dual nature of copper acting as a toxicant and a trace metal. We assume that if the cytoprotective properties of copper can still be observed at a dose of 0.2 mg per rat, then at a dose of 0.25 mg per rat its cytotoxic effects already predominate.

Further studies are needed to clarify the nature and dependency type of the dose–response relationships between size, concentration, and identity of NPs.

## 4. Materials and Methods

### 4.1. Synthesis of Nanoparticles

The suspensions of nanoparticles were obtained by laser ablation of 99.99% pure, 1 mm thick plates of elements in ca. 30 mL of deionized water. The Fmark-20RL laser system (LTC, Saint Petersburg, Russia) based on a Yb fiber laser with a wavelength of 1080 nm, pulse duration of 100 ns, pulse energy of 1 mJ, and repetition rate of 21 kHz was used. The target surface, cleaned by deionized water in an ultrasonic bath, was processed with a spot of laser irradiation with a fixed diameter of 40 μm and the fluence ranged from 15 to 80 J/cm^2^. The water layer thickness above the target ranged from 2 to 10 mm. The scanning velocity was about 270 mm/s, while the scanned area ranged from 25 to 300 mm^2^. A motor-driven agitator was used to reduce the scattering of the laser beam on cavitation bubbles and to remove the ablated nanoparticle cloud from the irradiated area during ablation. The ablation process lasted from 1 to 60 min. The value of the ablated mass was measured by weighing the target before and after ablation using a ME 235 S analytical balance (Sartorius AG, Göttingen, Germany).

This technique produced suspensions with a sufficiently narrow size distribution of nanoparticles (NPs) and although the singlet NPs tend to stick together, the resulting aggregates were usually loose and rather small (Figure 11).

All the element oxide NPs under study had a spherical shape. Their sizes and experimental doses are shown in Table 1.

The suspension stability was characterized by the value of Zeta potential measured by electrophoretic light scattering using a Zetasizer Nano ZS analyzer (Malvern Panalytical, Malvern, UK). A high suspension stability (Zeta potential up to 42 mV) allowed us to increase the concentration of NPs by partial water evaporation at 50 °C on a heat plate. The concentrations of the stable suspension were achieved without appreciable change in the size and chemical composition of the synthesized nanoparticles. The size distribution was measured by dynamic light scattering (DLS) using the Zetasizer Nano ZS analyzer and by a statistical analysis of the high-resolution images obtained using a CrossBeam Workstation Auriga scanning electron microscope (SEM) (Carl Zeiss, Jena, Germany). The results showed that the suspensions produced as above were highly stable and maintained their characteristics without increasing particle aggregation over periods sufficient for carrying out the animal experiments described below. No chemical stabilizers were added to either of the suspensions.

### 4.2. Experimental Animals

The experiment was conducted on outbred female albino rats aged 3.5 months and weighing about 200 g at the start. The animals were kept in a separate vivarium room of our center. They breathed unfiltered air and were given bottled artesian water and standard balanced feed. The rats were randomly assigned into 17 groups of 10 animals. We conducted 5 experiments: (1) CuO, PbO, CdO, Fe_2_O_3_, and NiO nanoparticles at a dose of 0.5 mg per animal; (2) SiO_2_ nanoparticles at a dose of 0.5 mg per animal; (3) NiO and Mn_3_O_4_ nanoparticles at a dose of 0.25 mg per animal; (4) CuO and SeO nanoparticles at a dose of 0.25 mg per animal; (5) CuO and PbO nanoparticles at a dose of 0.2 mg per animal. Each experiment had its own control group.

Treated animals were intratracheally administered with various doses of nanoparticles (1 mL). The control group received 1 mL deionized water.

The experiment was designed and implemented in accordance with the International Guiding Principles for Biomedical Research Involving Animals developed by the Council for International Organizations of Medical Sciences and the International Council for Laboratory Animal Science (2012), and was approved by the Ethics Committee of the Yekaterinburg Medical Research Center for Prophylaxis and Health Protection in Industrial Workers (Protocol No. 53 of 21 January 2014; Protocol No. 58 of 18 January 2016; Protocol No. 9 of 8 October 2018; Protocol No. 2 of 20 April 2020; and Protocol No. 2 of 20 April 2021).

### 4.3. Cytotoxicity Assessment

The cytotoxicity of nanoparticles was assessed by alveolar phagocytosis. Animals were injected intratracheally with 1 mL of a suspension of nanoparticles under ether round anesthesia. After 24 h, the bronchoalveolar lavage fluid was collected and a sample of the fluid with cells was obtained using a laboratory mixer. The cells were stained with methylene blue in 3% (*v*/*v*) acetic acid solution. We then counted the total number of cells in the aliquot using a light microscope in the Goryaev chamber with the following recalculation for the total volume of bronchoalveolar lavage fluid (BALF).

Then, 200 g of the BALF was centrifuged at 1000 r/min for 4 min and the supernatant was taken for biochemical testing of amylase, aspartate and alanine aminotransferase (AST, ALT), lactate dehydrogenase (LDH), and gamma-glutamyl transpeptidase (GGT) using a Cobas Integra 400 plus automated analyzer (Roche Diagnostics GmbH, Germany) with the appropriate test kits: alfa-Amylase EPS COBAS INTEGRA/cobas c system, AST IFCC COBAS INTEGRA/cobas c system, ALT IFCC COBAS INTEGRA/cobas c system, Lactate Dehydrogenase acc. to IFCC COBAS INTEGRA/cobas c system, and GGT IFCC COBAS INTEGRA/cobas c system (Roche Diagnostics GmbH, Switzerland). All the clinical laboratory BALF tests were performed using the techniques described in the many manuals [54].

The cell pellets were carefully resuspended in a small amount of the supernatant to prepare liquid-based preparations that were air dried, fixed in methanol, and stained with azure eosin methylene blue. At least 100 cells were counted on each smear using a light microscope while identifying alveolar macrophages, neutrophilic leukocytes, and eosinophils.

A single intratracheal instillation of 1 mL of NP water suspension (or of the same de-ionized water without particles for the controls) served as an experimental model for the response of the lower airways to particle deposition. The adequacy of this model for solving such problems is sufficient as it has been shown repeatedly that important qualitative and quantitative patterns of the response of the pulmonary free cell population (in particular, its dependence on the cytotoxicity of deposited particles) observed in inhalation exposures to dust particles are principally the same in the case of their intratracheal administration [17]. Animal experiments based on comparative inhalation exposures to nanoparticles and microparticles with known characteristics in equal concentrations would be, above any doubt, desirable. However, such experiments are associated with technical and financial difficulties that considerably limit research opportunities.

### 4.4. Statistical and Mathematical Analysis

The statistical significance of differences between the group arithmetic means was estimated using Student’s *t*-test with Bonferroni correction for multiple comparisons.

We normalized the data in the experimental groups to the same values in the corresponding control groups to compare parameters in different experiments, taking the control values as 1.

## 5. Conclusions

The experiments with single instillations of CuO, PbO, CdO, Fe_2_O_3_, NiO, SiO_2_, Mn_3_O_4_, and SeO nanoparticles at different doses (0 to 0.5 mg per animal) have shown the following.

The chemical nature of nanoparticles determines their cytotoxicity, probably due to increased oxidative stress, penetration of NPs into cells and their dissolution in biological fluids. At a dose of 0.5 mg per animal, the level of cytotoxicity declines in the following order: CuO NPs > PbO NPs > CdO NPs > NiO NPs > SiO_2_ NPs > Fe_2_O_3_ NPs. At a dose of 0.25 mg per animal, the cytotoxicity changes as follows: NiO NPs > Mn_3_O_4_ NPs > CuO NPs > SeO NPs.

Cytotoxicity demonstrates a non-linear increase with an increasing dose of the same chemical element (from 0 to 0.5 mg per animal). Thus, NPs of essential copper are likely to exert both protective and toxic effects while lead and nickel nanoparticles cause damage in all studied dose ranges.

A high level of intracellular enzymes (amylase, AST, ALT, and LDH) in the supernatant gives evidence of the cytotoxic effect of nanoparticles.

Alterations in the cytological parameters of the bronchoalveolar lavage and biochemical characteristics of the supernatant can be used to predict the danger of new nanomaterials based on their comparative assessment with the available tested samples of nanoparticles.

Finally, we should note that further studies are needed to clarify the nature and dependency type of the dose–response relationships between size, concentration, and identity of NPs.

### Study Limitations

The experiments were conducted at different times using animals of more than one livestock. Such physical characteristics of nanoparticles as solubility in water and biological fluids, charge, adsorption capacity, resistance to aggregation, hydrophobicity, adhesion to surfaces, and the ability to generate free radicals have not been studied. Extrapolation of data from rodents to humans shall be done with caution, since cytotoxicity has been characterized only based on the main cellular and biochemical parameters.

## Figures and Tables

**Figure 1 ijms-24-08383-f001:**
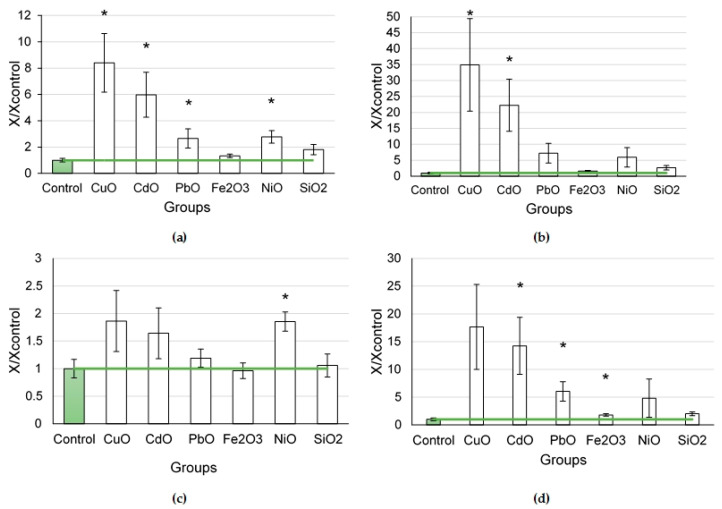
Cellular parameters of the bronchoalveolar lavage fluid following intratracheal instillation of element oxide nanoparticles at a dose of 0.5 mg per animal: (**a**) total cellularity; (**b**) number of neutrophilic leukocytes (NL); (**c**) number of alveolar macrophages (AM); (**d**) NL to AM ratio; the control is taken as 1, other values are given in relation to control. *—statistically different from the control groups (*p* < 0.05, based on Student’s *t*-test).

**Figure 2 ijms-24-08383-f002:**
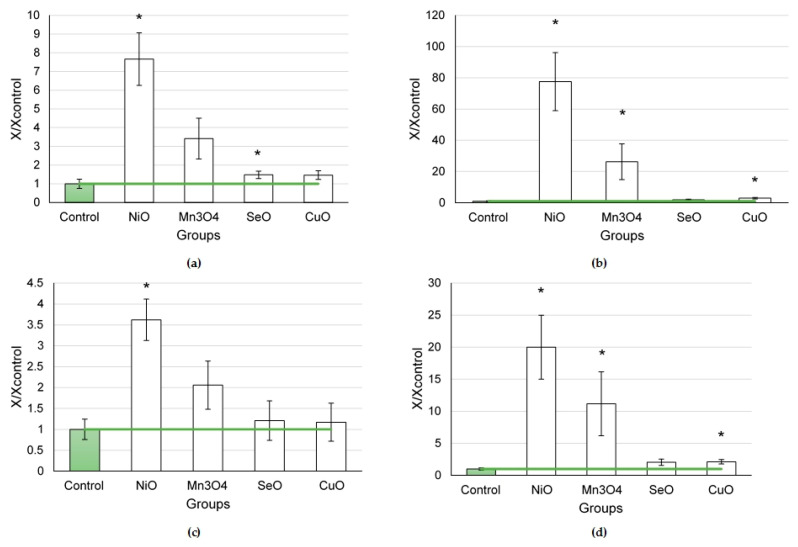
Cellular parameters of the bronchoalveolar lavage fluid following intratracheal instillation of element oxide nanoparticles at a dose of 0.25 mg per animal: (**a**) total cellularity; (**b**) number of neutrophilic leukocytes (NL); (**c**) number of alveolar macrophages (AM); (**d**) NL to AM ratio; the control is taken as 1, other values are given in relation to control. *—statistically different from the control groups (*p* < 0.05, based on Student’s *t*-test).

**Figure 3 ijms-24-08383-f003:**
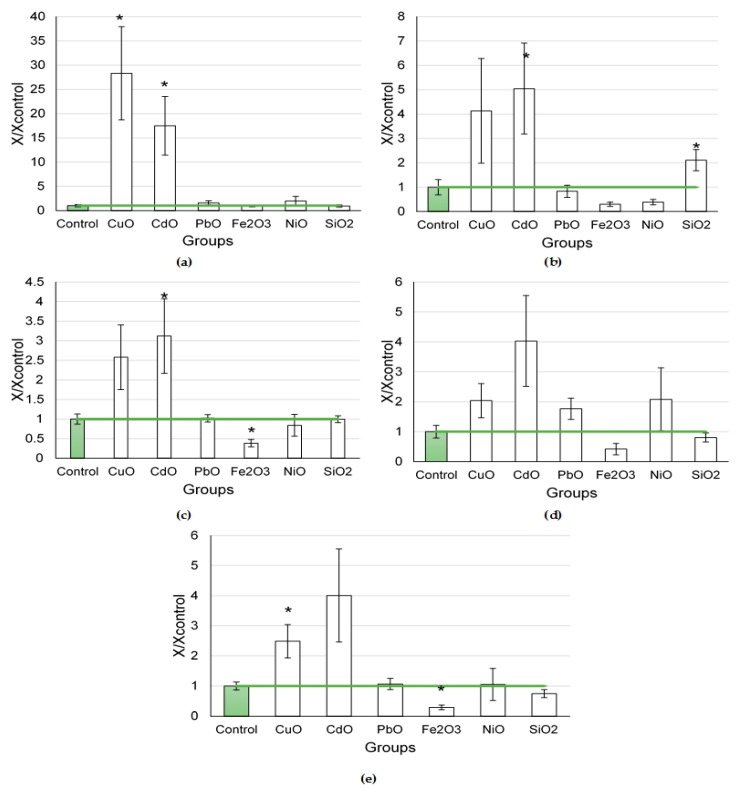
Biochemical parameters of the supernatant of bronchoalveolar lavage fluid following intratracheal instillation of element oxide nanoparticles at a dose of 0.5 mg per animal: (**a**) amylase; (**b**) ALT; (**c**) AST; (**d**) GGT; (**e**) LDH; the control is taken as 1, other values are given in relation to control. *—statistically different from the control groups (*p* < 0.05, based on Student’s *t*-test).

**Figure 4 ijms-24-08383-f004:**
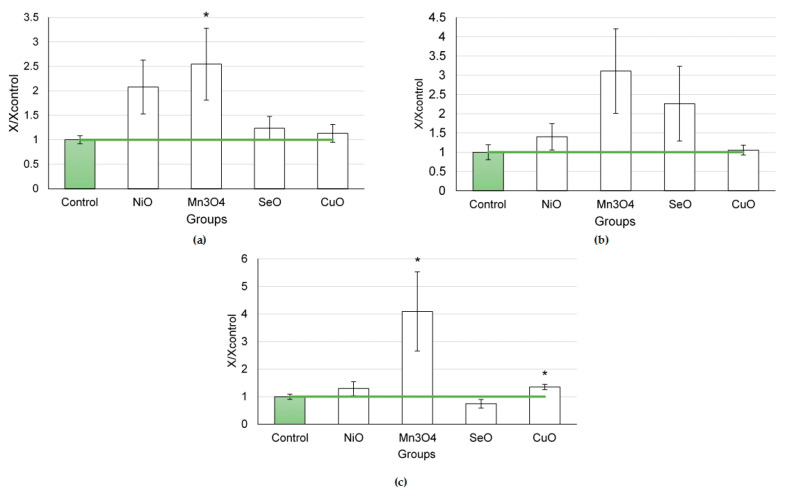
Biochemical parameters of the supernatant of bronchoalveolar lavage fluid following intratracheal instillation of element oxide nanoparticles at a dose of 0.25 mg per animal: (**a**) amylase; (**b**) GGT; (**c**) LDH; the control is taken as 1, other values are given in relation to control. *—statistically different from the control groups (*p* < 0.05, based on Student’s *t*-test).

**Figure 5 ijms-24-08383-f005:**
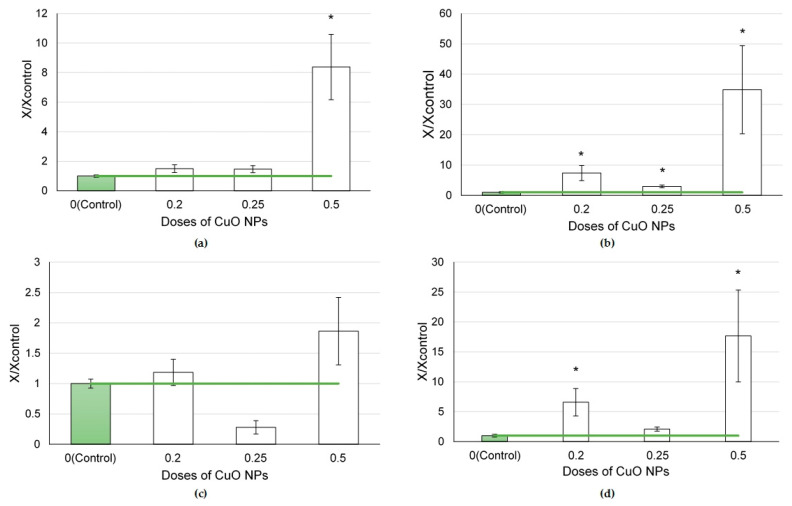
Cellular parameters of the bronchoalveolar lavage fluid following intratracheal instillation of copper oxide nanoparticles at doses of 0.2, 0.25 and 0.5 mg per animal: (**a**) total cellularity; (**b**) number of neutrophilic leukocytes (NL); (**c**) number of alveolar macrophages (AM); (**d**) NL to AM ratio; the control is taken as 1, other values are given in relation to control. *—statistically different from the control groups (*p* < 0.05, based on Student’s *t*-test).

**Figure 6 ijms-24-08383-f006:**
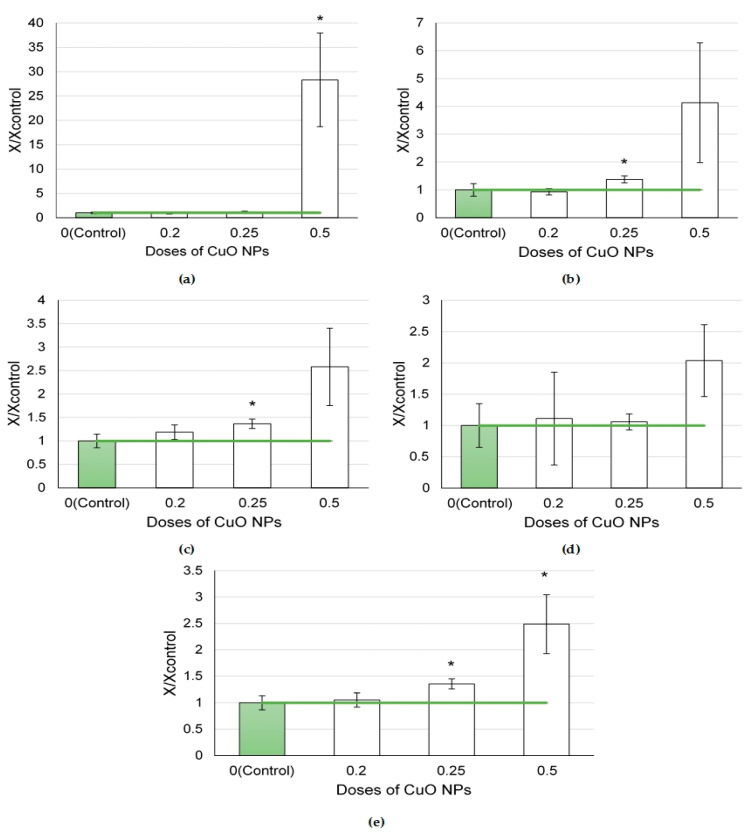
Biochemical parameters of the supernatant of bronchoalveolar lavage fluid following intratracheal instillation of copper oxide nanoparticles at doses of 0.2, 0.25, and 0.5 mg per animal: (**a**) amylase; (**b**) ALT; (**c**) AST; (**d**) GGT; (**e**) LDH; the control is taken as 1, other values are given in relation to control. *—statistically different from the control groups (*p* < 0.05, based on Student’s *t*-test).

**Figure 7 ijms-24-08383-f007:**
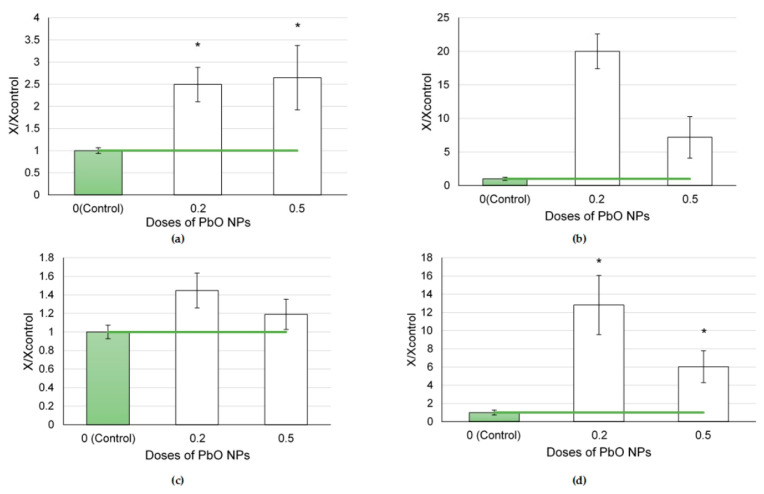
Cellular parameters of the bronchoalveolar lavage fluid following intratracheal instillation of lead oxide nanoparticles at doses of 0.2 and 0.5 mg per animal: (**a**) total cellularity; (**b**) number of neutrophilic leukocytes (NL); (**c**) number of alveolar macrophages (AM); (**d**) NL to AM ratio; the control is taken as 1, other values are given in relation to control. *—statistically different from the control groups (*p* < 0.05, based on Student’s *t*-test).

**Figure 8 ijms-24-08383-f008:**
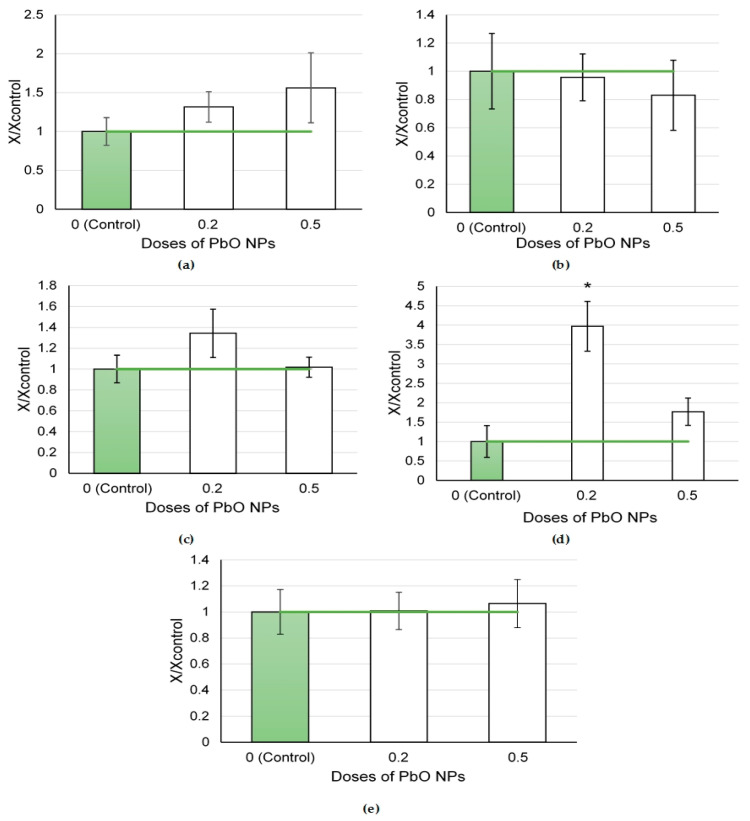
Biochemical parameters of the supernatant of bronchoalveolar lavage fluid following intratracheal instillation of lead oxide nanoparticles at doses of 0.2 and 0.5 mg per animal: (**a**) amylase; (**b**) ALT; (**c**) AST; (**d**) GGT; (**e**) LDH; the control is taken as 1, other values are given in relation to control. *—statistically different from the control groups (*p* < 0.05, based on Student’s *t*-test).

**Figure 9 ijms-24-08383-f009:**
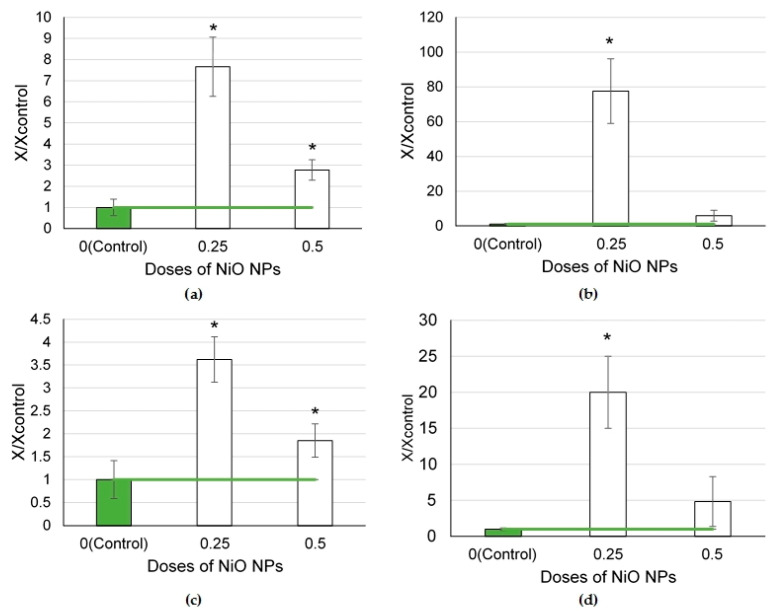
Cellular parameters of the bronchoalveolar lavage fluid following intratracheal instillation of nickel oxide nanoparticles at doses of 0.25 and 0.5 mg per animal: (**a**) total cellularity; (**b**) number of neutrophilic leukocytes (NL); (**c**) number of alveolar macrophages (AM); (**d**) NL to AM ratio; the control is taken as 1, other values are given in relation to control. *—statistically different from the control groups (*p* < 0.05, based on Student’s *t*-test).

**Figure 10 ijms-24-08383-f010:**
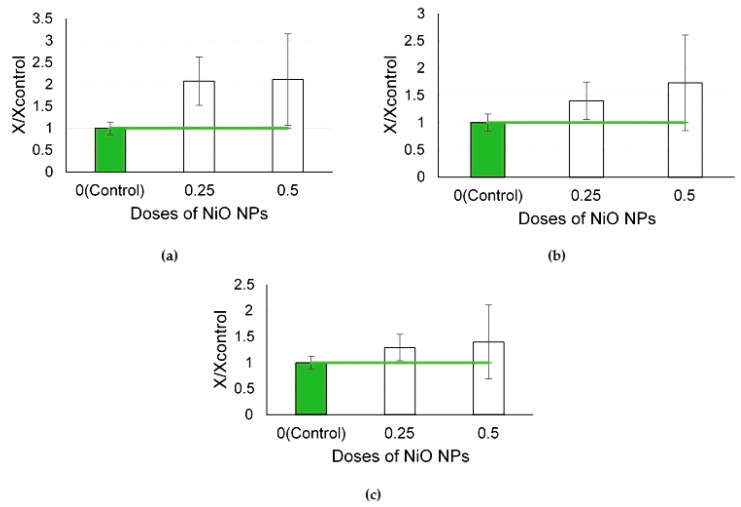
Biochemical parameters of the supernatant of bronchoalveolar lavage fluid following intratracheal instillation of nickel oxide nanoparticles at doses of 0.25 and 0.5 mg per animal: (**a**) amylase; (**b**) GGT; (**c**) LDH; the control is taken as 1, other values are given in relation to control.

**Figure 11 ijms-24-08383-f011:**
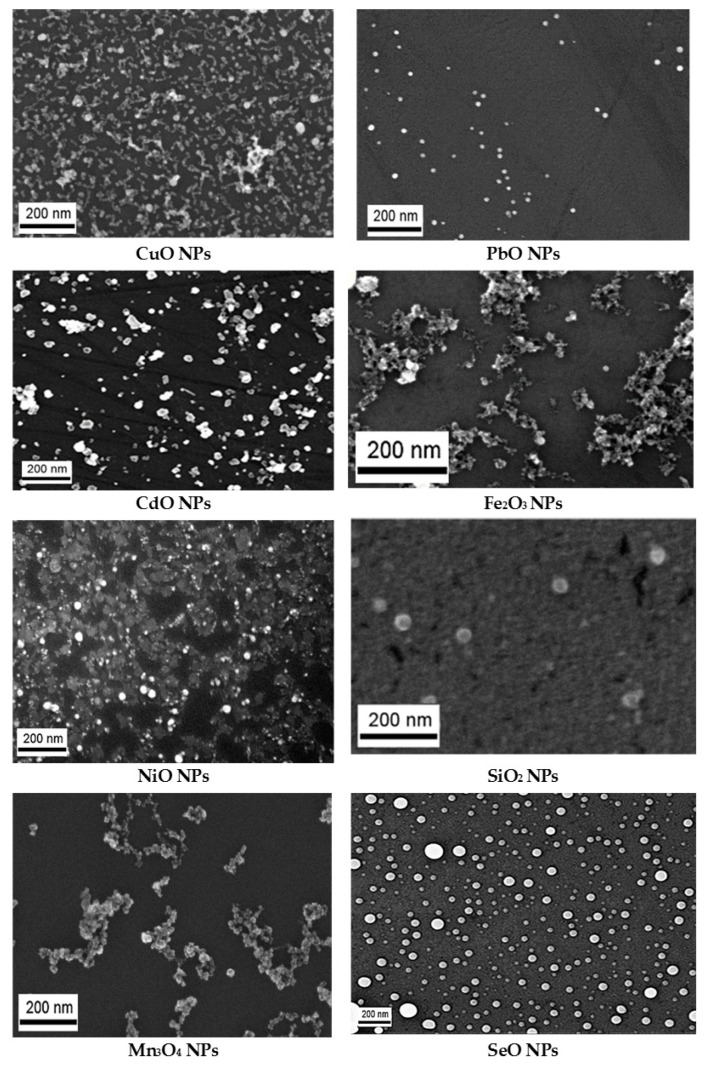
Nanoparticles of various chemical natures synthesized for animal experiments by laser ablation. Images of nanoparticles obtained by scanning electron microscopy at ×30,000.

**Table 1 ijms-24-08383-t001:** Size and doses of nanoparticles.

Nanoparticles	Size, nm	Doses, mg/mL
CuO	24.5 ± 4.8	0.2
	21.0 ± 4.0	0.25
	21.0 ± 4.0	0.5
PbO	47.0 ± 16.0	0.2
	23.0 ± 5.0	0.5
CdO	65.0 ± 16.0	0.5
Fe_2_O_3_	18.0 ± 4.0	0.5
NiO	16.7 ± 8.2	0.25
	16.7 ± 8.2	0.5
SiO_2_	43.0 ± 11.0	0.5
Mn_3_O_4_	18.4 ± 5.4	0.25
SeO	51.0 ± 14.0	0.25

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
