# Peer review of "Comparative Evaluation of the Cytotoxic Effects of Metal Oxide and Metalloid Oxide Nanoparticles: An Experimental Study"

_ijms, 2023, doi:10.3390/ijms24098383_

Round 1
Reviewer 1 Report
Title: I propose to modify the title: “Comparative Evaluation of cytotoxic effects of Nanoparticles of Oxides of Metals and Metalloids: An Experimental Study” Not the metalloids but their oxides are evaluated,
General Comments:
Line 135: It is remarkable that the results of copper and lead oxides are presented and discussed in detail but not the tremendous dependency on doses with NiO. This should be added.
Line 334. Table 1: Whereas the size of the nanoparticles is nearly) the same for different doses with CuO and NiO, the sizes for PbO differ significantly. Toxic effects may depend on particle size / surface area. What does this mean regarding the discussion of dependency of the effects on PbO doses (see section 2.2.2 and lines 210-313)
Lines 362-363: How many animals are treated per dose? Can you exclude that the variations between different doses are originated by different responses of individual animals?
Lines 398-399: It is stated the cytotoxicity is determined by the chemical identity of the particles and their dissolution behavior. This cannot be inferred by discussion section above where the influences of their essentiality and redox behavior are explained.
Lines 409-411: Considering the non-linear relationships it is not clear, how the method can be used for screening purposes. What follows with regard to protective measures at the workplace?
Minor comments:
Lines 242-244: It is referred to reference 38: I suppose that SiO2 is meant. Is it amorphous or crystalline.
Author Response
Dear Reviewer,
Thank you for reviewing our manuscript and for your questions and valuable comments used to improve it.
The title of the article was changed.
Line 135 (Line 151 in the latest version of the article):
We added results of nickel oxide nanoparticles in section 2.2.3 and Discussion (section 3.2).
Line 334 (Line 398 in the latest version of the article):
Larger PbO NPs could be easily recognized by defense systems of the body and phagocytosed than smaller [Sukhanova, A.; Bozrova, S.; Sokolov, P.; Berestovoy, M.; Karaulov, A.; Nabiev I. Dependence of nanoparticle toxicity on their physical and chemical properties. Nanoscale Res. Lett. 2018, 13(1), 44. https://doi.org/10.1186/s11671-018-2457-x].
Investigating the influence of nanoparticle sizes on their cytotoxicity is beyond the scope of the article. We plan to continue these studies in the future.
Line 362-363 (Line 433-434 in the latest version of the article):
We added explanation about animals in section 4.2:
The rats were randomly assigned into 17 groups of ten animals. We conducted 5 experiments: 1) CuO, PbO, CdO, Fe2O3 and NiO nanoparticles in dose 0.5 mg per animal; 2) SiO2 nanoparticles nanoparticles in dose 0.5 mg per animal; 3) NiO and Mn3O4 nanoparticles in dose 0.25 mg per animal; 4) CuO and SeO nanoparticles in dose 0.25 mg per animal; 5) CuO and PbO nanoparticles in dose 0.2 mg per animal. Each experiment had its own control group.
There are intra-interspecific and genetic differences in the body's response to various influences. To align them, we used statistical analysis.
Line 398-399 (Line 476-477 in the latest version of the article):
Thank you for your comment! We have extended the output.
Lines 409-411 (Line 489-493 in the latest version of the article):
The word "screening" has been removed. New version:
Alterations in cytological parameters of the bronchoalveolar lavage and biochemical characteristics of the supernatant can be used to predict the danger of new nanomaterials based on their comparative assessment with the available tested samples of nanoparticles. Protective measures for workers can be the development of safe levels of nanoparticles and increasing the body's resistance to their harmful cytotoxic effects.
Lines 242-244 (Line 292-294 in the latest version of the article):
Unfortunately, the authors [38] did not specify in their work what kind of silicon was in the nanoparticles. As it is not indicated on the manufacturer's website (Sigma). Most likely, amorphous silicon dioxide nanoparticles were meant.
Reviewer 2 Report
Here are some comments and remarks to the authors :
Title 2.1.1 : Explain BALF (1st time seen in text), other explanation come p.13
Figure1 : add the signification of the * in the graph. On how many animals this experiment was conducted ? This is never mentioned in the paper.
Figure 9 : add explanation on the way these picture were obtained, and about the zoom used.
Line 108 : add that this is compared to the effect of Mn3O4 (not really clear) and 3 times highr compared to control (If I have well understood).
Lines 126-127 : this seems not true for SeO and CuO based on the figure 4. Maybe modify slightly the sentence to take this into account.
Figure 3 and 4 : How is measured the enzymatic activity metionned in the figures ? I do not see any description of the method in the paper, please add something about this.
How the authors takes into account different response time to intratracheal instillation of NPs oxydes ? Can this response time have an impact on the NL/AM ratio ? What would be the resultats after 48h or 72h and why the authors did not test this ?
Paragraph 3.2 : Dose-response are often non linear and authors tested 3 different concentrations of NPs oxydes ? Why not testing more concentrations to confirm the non linearity of the dose response relation ? I mean 3 points is the very minimum to prove a non linearity.
To be able to judge about the veracity of the non effect at 0,25 mg, but effect at 0,2 mg and 0,5 mg, some further details would be very useful, like the number of animals used and how many times this experiment was conducted ?
Author Response
Dear Reviewer,
We appreciate your detailed review and have done our best to improve the manuscript in accordance with your valuable comments and recommendations.
In section “4.2. Experimental animals” information about the number of animals and groups was added. Description of the methods to measure the enzymatic activity was added in section “4.3. Cytotoxicity assessment”.
We added the signification of the * in the Figures. Explanations added to Figure 9 (11 in the latest version of the article).
Line 108 (Lines 117-118 in the latest version of the article):
Changed.
Lines 126-127 (Lines 138-139 in the latest version of the article):
Done.
About exposure time:
Previous experiments have studied the dynamics of mobilization of the cellular composition obtained during bronchoalveolar lavage (BAL) in the experiment.
It was found that exactly 24 hours after the intratracheal injection of particles, cell mobilization was maximal in comparison with shorter periods (hours) or larger (days) [Werb, Z. Respiratory Defense Mechanisms (In Two Parts). Part I and Part II. Lung Biology in Health and Disease. Eds: J.D. Brain, D.F. Proctor, L.M. Reid. Marcel Dekker, New York. 1978, 1216 pp.]
Therefore, we evaluate the cytological characteristics of BAL 24 hours after the instillation of NPs, that is, at peak levels.
About more doses:
We completely agree with the opinion of the reviewer. We have added the following to the discussion and conclusions:
Further studies are needed to clarify the nature and dependency type of the dose-response relationships between size, concentration, identity of NPs.
Round 2
Reviewer 1 Report
Major improvements of the manuscript. Thank you.
Reviewer 2 Report
Dear Authors,
Thak you for taking into account all my previous comments in this new version. The paper is fine for me by now.
Best